# Physiological Performance and Grain Yield Components of Common Buckwheat (*Fagopyrum esculentum* Moench) Cultivated Under Different N Rates

**DOI:** 10.3390/plants14132037

**Published:** 2025-07-03

**Authors:** Jorge González-Villagra, Jaime Solano, Kevin Ávila, Jaime Tranamil-Manquein, Ricardo Tighe-Neira, Alejandra Ribera-Fonseca, Claudio Inostroza-Blancheteau

**Affiliations:** 1Escuela de Agronomía, Facultad de Ciencias, Ingeniería y Tecnología, Universidad Mayor, Temuco 4801043, Chile; 2Centro para la Resiliencia, Adaptación y Mitigación (CReAM), Universidad Mayor, Av. Alemania 281, Temuco 4801043, Chile; 3Departamento de Ciencias Agropecuarias y Acuícolas, Facultad de Recursos Naturales, Universidad Católica de Temuco, Temuco P.O. Box 15-D, Chile; jsolano@uct.cl (J.S.); oavila@uct.cl (K.Á.); jtranamil@uct.cl (J.T.-M.); rtighe@uct.cl (R.T.-N.); claudio.inostroza@uct.cl (C.I.-B.); 4Núcleo de Investigación en Producción Alimentaria, Facultad de Recursos Naturales, Universidad Católica de Temuco, Temuco P.O. Box 15-D, Chile; 5Departamento de Producción Agropecuaria, Facultad de Ciencias Agropecuarias y Medioambiente, Universidad de La Frontera, Temuco P.O. Box 24-D, Chile; alejandra.ribera@ufrontera.cl; 6Millenium Nucleus in Data Science for Plant Resilience (Phytolearning), Santiago 8370186, Chile

**Keywords:** CO_2_ assimilation, stomatal conductance, crop yield, shoot diameter

## Abstract

Buckwheat (*Fagopyrum esculentum* Moech) is a “gluten-free” pseudocereal with high-quality proteins and human health properties, increasing its cultivation worldwide. However, the role of nitrogen (N) in plant growth and yield components has received little attention in buckwheat. This study evaluated N’s effect on plant traits, photosynthetic performance, and grain yield components in buckwheat under field conditions. For this, Buckwheat cv. “Mancan” seeds were sown using five N rates: 0, 30, 45, 60, and 90 kg N ha^−1^. Then, physiological performance and grain yield components were evaluated at harvest. Our study revealed that buckwheat plants subjected to 0 and 30 kg N ha^−1^ showed the greatest chlorophyll fluorescence *a* parameters including maximum quantum yield of PSII (Fv′/Fm′), effective quantum yield of PSII (ФPSII), and electron transport rate (ETR) among N treatments; meanwhile, at higher N rates (60 and 90 kg N ha^−1^), these parameters decayed. Similarly, plants treated with 90 kg N ha^−1^ showed the lowest CO_2_ assimilation among N treatments. In general, stomatal conductance (*g_s_*), transpiration (*E*), and intrinsic water use efficiency (WUE*_i_*) showed no significant changes among N treatments, with the exception of 30 kg N ha^−1^, which exhibited the highest WUE*_i_*. Concerning plant traits, plants grown under 60 and 90 kg N ha^−1^ exhibited the greatest plant height, number of branches, shoot biomass, and internode per plant among N treatments. By contrast, 30 kg N ha^−1^ showed the highest grain number, yield per plant, and grain yield among N treatments in *F. esculentum* plants. Based on the physiological and productive parameters, *F. esculentum* seems to have a low N requirement, exhibiting better results under the lowest N rates (30 kg N ha^−1^). Therefore, *F. esculentum* could be considered as an alternative for gluten-free food production with low N requirements in agricultural systems of southern Chile. Nonetheless, more studies are required to understand the effect of N biochemical and molecular regulation on plant traits and grain yield components in buckwheat.

## 1. Introduction

Common buckwheat (*Fagopyrum esculentum* Moench) is a re-emergent “gluten-free” pseudocereal with high-quality proteins, minerals, vitamins, and phenolic compounds, providing human health benefits, being suitable for people with celiac disease [1,2,3]. It has been reported that rutin is among the most important molecules found in buckwheat seeds, preventing diabetes, hypertension, and liver damage [4,5]. This crop was very popular during the 19th century, decreasing its cultivation due to the expansion of wheat in Europe [6]. Therefore, this crop has received less attention in agronomical and physiological studies [7,8]. Currently, common buckwheat cultivation is low (less area) compared to high-yielding crops such as maize, rice, and wheat [9]. According to FAO STAT [9], a total of 1,988,545 ha was sown with buckwheat worldwide, producing 1,875,067 tons during the 2021 season, with Russia, China, and France being the main grower countries. Interestingly, buckwheat is considered a low-input species with weed suppression activity due to rapid emergence and fast growth, resource competition, and allelopathic effects against weeds, which reduces the use of synthetic herbicides [3,10,11]. On the other hand, some studies have reported that buckwheat is a melliferous species due to its nectar composition, which mainly contains glucose, fructose, sucrose, vitamins, and amino acids, attracting several pollinator insects such as *Apis mellifera*, *Apis cerana*, *Bombus* spp., *Andrena* spp., and *Diptera* species [12,13]. Therefore, common buckwheat is an interesting crop with several positive benefits to human health and the environment, which can contribute to agricultural sustainability for the next years. However, to increase its cultivation, more studies are needed to deepen agronomic, physiological, and yielding aspects. Currently, some studies have contributed to a better understanding of buckwheat growth and cultivation. For example, Arduini et al. [14] developed the first growth scale to follow growth and to predict harvest time in buckwheat. Other studies have reported the agronomical and physiological mechanism in response to sowing time, planting density, nutrient uptake, growth, and grain yield in buckwheat [11,15,16,17]. However, according to Rotili et al. [18] and Wan et al. [19], nitrogen’s (N’s) role in grain yield components has received little attention in buckwheat.

Nitrogen (N) is an important nutrient forming part of protein and nucleic acids and is involved in several physiological processes such as leaf development, photosynthesis, and source–sink dynamics [20,21]. It is well known that N deficit decreases plant growth and photosynthesis in several crops such as wheat, maize, barley, and rice; therefore, high N fertilization rates have been applied to maintain and/or increase crop yields, triggering global environmental issues [22]. However, little is known about N’s involvement in physiological and agronomic yield characteristics in buckwheat [18,23]. Thus, consequently, this study evaluated N’s role in determining plant traits, photosynthetic performance, and grain yield components in buckwheat under field conditions.

## 2. Results

### 2.1. Environmental Conditions

During the field assay, nineteen rainfall events were recorded between November and March in the 2019/2020 season (Figure 1). The accumulated rainfall reached 81.3 mm between sowing and harvest. The average maximum temperature was 23.8 °C during the experiment, with some days reaching between 30 and 35 °C during February (Figure 1). The average minimum temperature was 6.6 °C, with a few days ranging between 0 and 3 °C.

### 2.2. Chlorophyll Fluorescence and Gas Exchange Parameters

Our results revealed that the maximum photochemical efficiency of PSII (Fv′/Fm′) significantly decreased in 45, 60, and 90 kg N ha^−1^ treatments (with no changes among them) compared to the control (0 kg N ha^−1^) and 30 kg N ha^−1^ in *F. esculentum* (Figure 2A). Otherwise, the effective quantum yield of PSII (ФPSII) was significantly reduced in all N treatments, where 60 and 90 kg N ha^−1^ showed significant reductions (by about 11 and 25%, respectively) compared to the control treatment (Figure 2B). A similar effect was exhibited in the electron transport rate (ETR), which was reduced in all N treatments (Figure 2C). However, the lowest ETR values were observed in 60 and 90 kg N ha^−1^, decreasing around 10.3 and 16.6% compared to the control treatment. Regarding gas exchange parameters, our experiment revealed that CO_2_ assimilation showed a slight reduction in 30, 45, and 60 kg N ha^−1^. Meanwhile, a greater reduction (by 14%) was observed in the 90 kg N ha^−1^ treatment with respect to the control treatment (Figure 3A). Interestingly, stomatal conductance (*g_s_*) was reduced by 11% in 30 kg N ha^−1^ compared to 0 kg N ha^−1^ (Figure 3B). By contrast, no changes were observed in g_s_ among 0, 45, 60, and 90 kg N ha^−1^ treatments. A similar tendency was observed in transpiration (*E*), where 30 kg N ha^−1^ showed a slight reduction of 8.2% compared to the control N treatment in *F. esculentum* (Figure 3C). Additionally, *E* did not vary among 0, 45, 60, and 90 kg N ha^−1^. Interestingly, 30 kg N ha^−1^ exhibited the highest intrinsic water use efficiency (WUE*_i_*) level among the N treatments, and it was 9.8% greater with respect to the control and 22.8% higher compared to 90 kg N ha^−1^ (Figure 3D).

### 2.3. Plant Traits and Grain Yield Components

Our results revealed that plants subjected to higher N treatments (60 and 90 kg N ha^−1^) exhibited greater plant height (about 8%) compared to control plants (Table 1). Likewise, plants under 60 and 90 kg N ha^−1^ treatments showed the highest number of branches and internodes per plant. By contrast, shoot diameter did not vary among the different N treatments in *F. esculentum* plants. On the other hand, plants subjected to 60 and 90 kg N ha^−1^ treatments displayed the highest shoot biomass, which was related to higher plant height and branch and internode numbers (Table 1). Regarding the grain yield component, plants under 30 kg N ha^−1^ treatment had a higher grain number (68.6%) and yield per plant (27%) compared to control *F. esculentum* plants (Table 2). A similar tendency was observed in crop yield, where plants under 30 kg N ha^−1^ exhibited the highest yield, which was 27.8% greater with respect to plants under the control treatment. Interestingly, *F. esculentum* plants under the 30 kg N ha^−1^ treatment revealed greater grain number (79%), yield per plant (82.7%), and crop yield (45%) parameters compared to plants subjected to the 90 kg N ha^−1^ treatment. Although no changes were observed in the 1000-grain weight parameter among N treatments, a slight tendency to increase in 30 kg N ha^−1^ was observed.

## 3. Discussion

Common buckwheat (*Fagopyrum esculentum* M.) is a pseudocereal with beneficial properties for human health and the environment [3,11]. However, little is known about N’s effect on physiological and agronomic yield characteristics in *F. esculentum*. Thus, we investigated the role of N in determining plant traits, photosynthetic performance, and grain yield components in common buckwheat cv. “Mancan”. Several studies have reported that N is directly related with photosynthetic performance, plant growth, and crop yield in different crop species such as *Triticum aestivum, Oryza sativa, Gossypium hirsutum*, and *Zea mays* [24,25,26,27]. However, our study revealed that *F. esculetum* plants subjected to 0 and 30 kg N ha^−1^ treatments showed higher photosynthetic performance (including CO_2_ assimilation and chlorophyll fluorescence *a* parameters) compared to higher N rates, mainly in 90 kg N ha^−1^. Similar results were reported by Fang et al. [28], where higher photosynthesis was observed between 0 and 45 kg N ha^−1^, while higher N rates (between 45 and 90 kg ha^−1^) decreased photosynthetic activity, which was concomitant with lower yield under high N levels. Some authors reported that a reduction in photosynthetic activity could be explained by a decrease in intercellular CO_2_ concentration (*Ci*) dependent on stomatal limitations (stomatal closure) and/or due to biochemical limitations [28,29]. However, we observed that stomatal conductance showed no differences among N treatments, with the exception of 30 kg N ha^−1^, which showed a slight decrease. Interestingly, Fv′/Fm′, ФPSII, and ETR significantly decreased concomitant with higher N rates, which could be related with leaf shading triggered by greater plant heigh, mainly in 60 and 90 kg N ha^−1^, which was related to an increase in branch number, internode number, and shoot biomass (Table 1). Therefore, in our study, high N levels promoted greater plant biomass, decreasing chlorophyll fluorescence *a* parameters, which in turn decreased CO_2_ assimilation with no changes in stomatal conductance. Regarding grain yield components, similar yields were reported by Hornyák et al. [30], around 1000 kg ha^−1^, and Solano et al. [31], where the authors showed 1000 kg ha^−1^ in *F. esculentum* cultivated in Southern Chile. We observed that 30 kg N ha^−1^ showed the greatest crop yield and yield per plant, which was 32% greater compared to 90 kg N ha^−1^. The lower crop yield and yield per plant in 60 and 90 kg N ha^−1^ could be explained by low CO_2_ assimilation and lower chlorophyll fluorescence *a* parameters in these treatments. On the other hand, the higher crop yield and yield per plant in 30 kg N ha^−1^ were related to higher grain per plant, with no changes in 1000-grain weight among N treatments. Similar results were reported by Rotili et al. [18], who reported that *F. esculentum* showed a tendency to increase crop yield at low N rates. Thus, *F. esculentum* seems to be a promising species, being considered a low-nitrogen-demand crop. Interestingly, we observed that *F. esculentum* growth produced high yields in low available soil phosphorus (P) (18 mg kg^−1^), which could be an interesting topic of research, considering the dwindling global phosphate rock reserve and the increasing price of P fertilizer. In fact, Zhou et al. [32] reported that *F. esculentum* exhibits high efficiency in P uptake and grows in under-fertilized soils. Thus, it has been reported that F. esculentum is a well-known P-efficient crop due to its organic acid exudation by the roots, secretion of phosphatase for P acquisition, and low requirement for growth [33,34]. Therefore, *F. esculentum* is an interesting species, which could be cultivated in soils with low nitrogen and phosphorus availability.

## 4. Materials and Methods

### 4.1. Description of the Study Site and Weather Conditions

The field assay was performed at the Agricultural Experimental Station (38°39′ S, 72°27′ W) of the Universidad Católica de Temuco, located in Lautaro, La Araucanía Region, Chile, during the 2019/2020 season. The soil is classified as the Temuco series (Andisol, Typic Hapludands) [35]. The soil nutrient analysis showed a pH 5.6, 17.21% of organic matter, 30.5 mg kg^−1^ of N, 18 mg kg^−1^ of available P, 115 mg kg^−1^ of available K, and 1.2% Al saturation. Weather conditions were obtained from the Research Center INIA “Carillanca” and its Automatic Weather Station (AWS) located at a site 10 km from the experimental site (https://agrometeorologia.cl, accessed on 10 June 2024). The temperatures and rainfall data are shown in Figure 1.

### 4.2. Plant Material and Experimental Conditions

Prior to sowing, soil was plowed twice with a disk harrow, followed by one plowing with a vibro-cultivator. The experimental design was a randomized complete block with five N treatments and three replicates, giving a total of 15 plots. Each experimental plot was 6 m^2^ (3 m × 2 m). Buckwheat (*Fagopyrum esculentum* M.) cv. “Mancan” was sown manually at a rate of 30 kg ha^−1^, with 20 cm between rows and 5 cm between seeds within rows on 18 November 2019. Each plot was fertilized upon being sown with 45 kg ha^−1^ P_2_O_5_ and 65 kg ha^−1^ K_2_O as basal fertilization. Five N treatments were applied: 0 kg N ha^−1^ (as a control treatment), 30 kg N ha^−1^, 45 kg N ha^−1^, 60 kg N ha^−1^, and 90 kg N ha^−1^ [18]. Commercial granular urea (N, 46%) was used as N fertilizer. The N rates were applied at buckwheat sowing. At physiological maturity, buckwheat was harvested on 12 March 2020. Weeds, pests, and diseases were not observed in the field experiment; therefore, chemical control was not necessary.

### 4.3. Photosynthetic Performance

Photosynthetic performance was determined through the photochemical efficiency of PSII and gas exchange at the beginning of the anthesis stage (3 January 2020) as reported by Fang et al. [28] and Wang et al. [36] using an LI-6400XT portable gas exchange system (LI-COR Inc. Lincoln, NE, USA) at a temperature of 20 °C, 60% external humidity, 400 ppm CO_2_ concentration, and light intensity of 1000 µmol m^−2^ s^−1^ as described by Fang et al. [28]. The photochemical efficiency of PSII was determined by the in vivo fluorescence emission of chlorophyll *a*. The portable gas analyzer was equipped with a camera for fluorescence measurement (Fluorescence Li-6400-40, LI-COR Inc., Lincoln, NE, USA). The maximum photochemical efficiency of PSII (Fv′/Fm′), effective quantum yield of PSII (ФPSII), and electron transport rate (ETR) were determined with Equations (1)–(3) [37] as described by Reyes-Díaz et al. [38]. Gas exchange was determined measuring the CO_2_ assimilation, stomatal conductance, and transpiration following the protocol of Reyes-Díaz et al. [38]. Intrinsic water use efficiency (WUE*_i_*) was determined as the ratio between CO_2_ assimilation and stomatal conductance (*g_s_*) as reported by Navarrete-Campos et al. [39]. Photosynthetic performance was determined in fully expanded leaves from the top of five plants per plot between 08:00 and 10:00 h on a sunny day.Fv′/Fm′ = (Fm′ − Fo′)/Fm′(1)ФPSII = (Fm′ − Ft′)/Fm′(2)ETR = PPF × 0.5 × ФPSII × 0.84(3)

### 4.4. Morphological Traits and Grain Yield Components

Ten plants from each plot were randomly selected to determine plant traits such as height, stem diameter, branch number, internode number, and shoot biomass. Stem diameter was determined using a digital caliper at the middle of the basal 2nd internode. For grain yield components, the grain number per plant, yield per plant, 1000-grain weight, and grain yield were determined from 1 m^2^ (in the middle rows of each plot) as previously described Fang et al. [28], using an analytical balance (Model BA2204B, Biobase Meihua Trading, Jinan, China). Seeds were air-dried for two weeks before yield analysis.

### 4.5. Statistical Analyses

To evaluate the effect of nitrogen (N) fertilization on physiological performance, plant traits, and grain yield components in *F. esculentum*, we performed a series of statistical analyses using SigmaStat version 3.5 (SPSS Inc., Chicago, IL, USA). All data were first tested for assumptions of normality and homoscedasticity using the Kolmogorov–Smirnov and Levene tests, respectively. A one-way analysis of variance (ANOVA) was conducted to determine the statistical significance of differences among five N rates (0, 30, 45, 60, and 90 kg N ha^−1^) on individual response variables, including chlorophyll fluorescence parameters (Fv′/Fm′, ФPSII, and ETR), gas exchange variables (CO_2_ assimilation, stomatal conductance, transpiration, and WUEi), morphological traits (plant height, shoot diameter, number of branches, internodes, and biomass), and yield components (grain number per plant, yield per plant, 1000-grain weight, and total crop yield). Post hoc comparisons were performed using Tukey’s Honest Significant Difference (HSD) test to detect pairwise differences among N treatments at a significance level of *p* < 0.05. All physiological measurements were collected from five biological replicates per plot for fluorescence and gas exchange parameters, and ten plants per plot were evaluated for plant morphology and yield parameters. Data are represented as means ± standard errors (SE) and were constructed for all measured variables.

## 5. Conclusions

It is well known that *F. esculentum* is a pseudocereal with high-quality proteins and health-promoting properties. However, the role of nitrogen (N) in regulating its physiological traits and yield components has been poorly explored. Thus, this study provides new insights into the physiological and agronomic responses of *Fagopyrum esculentum* to varying nitrogen (N) fertilization rates under field conditions. The field study demonstrated that excessive N application (60–90 kg N ha^−1^) increased vegetative growth but negatively impacted photosynthetic performance—particularly chlorophyll fluorescence parameters and CO_2_ assimilation—likely due to self-shading effects. In contrast, the application of 30 kg N ha^−1^ significantly improved grain yield and yield per plant, primarily by increasing grain number rather than grain weight. Notably, this yield enhancement occurred even in soils with high organic matter but limited phosphorus availability. These findings suggest that *F. esculentum* has a low nitrogen requirement and may be a suitable crop for low-input agricultural systems aimed at reducing synthetic N fertilizer use. However, further research is necessary to elucidate the biochemical and molecular mechanisms by which nitrogen influences plant development and yield formation in buckwheat.

## Figures and Tables

**Figure 1 plants-14-02037-f001:**
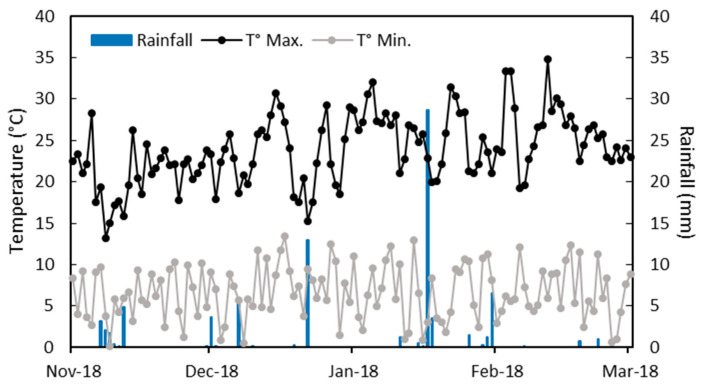
Daily values for maximum (T° Max.) and minimum (T° Min.) temperatures, and rainfall between sowing and harvest of F. esculentum.

**Figure 2 plants-14-02037-f002:**
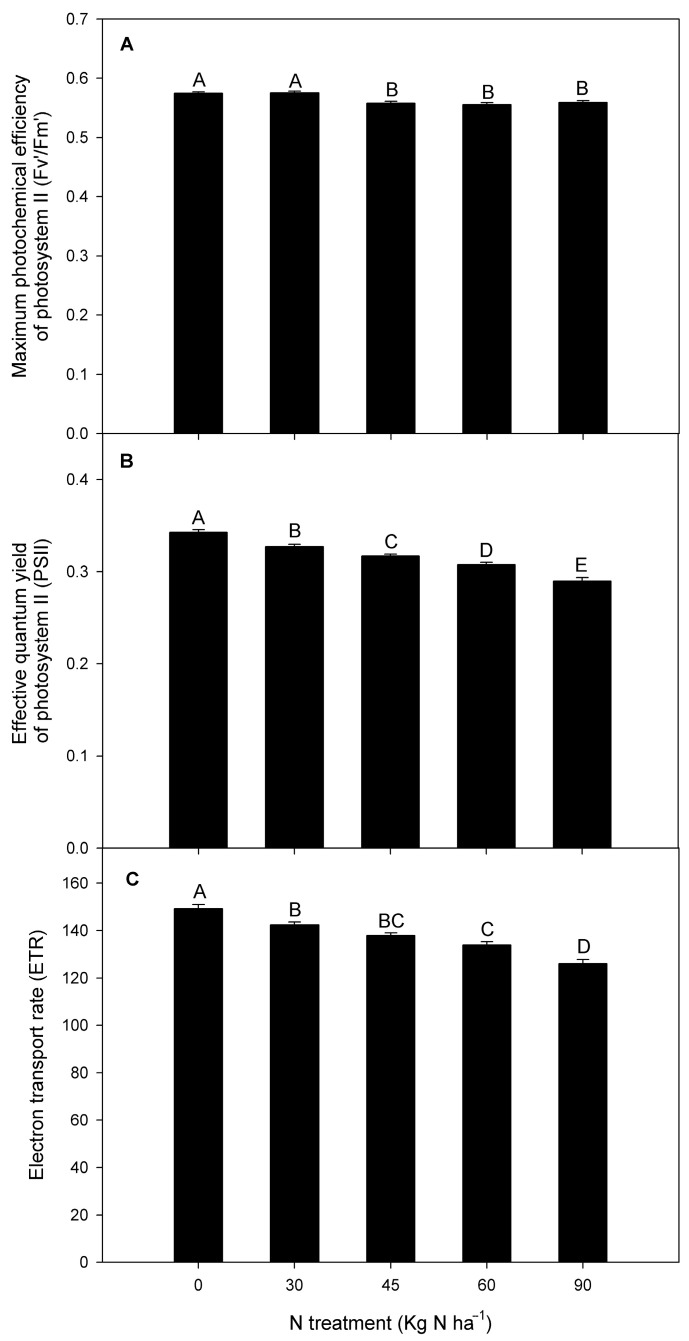
Chlorophyll fluorescence parameters. (**A**) Maximum photochemical efficiency of photosystem II in light-adapted leaves (Fv′/Fm′), (**B**) effective quantum yield of photosystem II (ФPSII), and (**C**) electron transport rate (ETR) in buckwheat (*F. esculentum*) subjected to nitrogen treatments (0, 30, 45, 60, and 90 kg N ha^−1^). Different uppercase letters indicate significant difference among treatments according to Tukey’s test (*p* < 0.05). The data are means ± SE (n = 5).

**Figure 3 plants-14-02037-f003:**
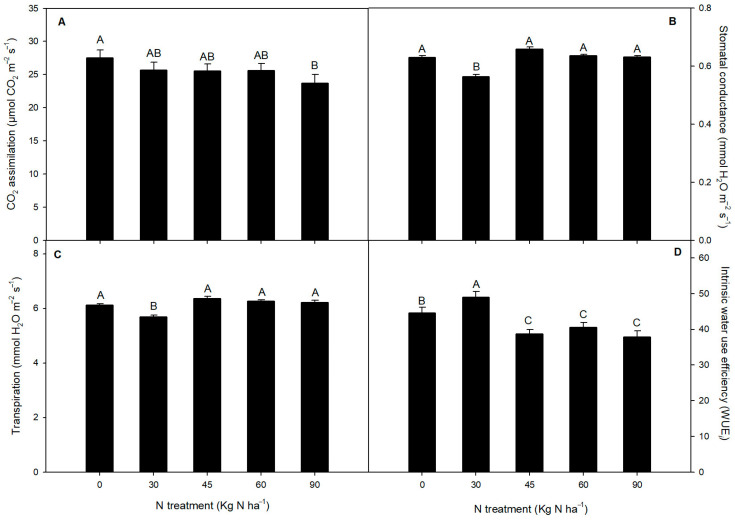
Gas exchange. (**A**) CO_2_ assimilation (*P_n_*), (**B**) stomatal conductance (*g_s_*), (**C**) transpiration (*E*), and (**D**) intrinsic water use efficiency (WUE*_i_*) in buckwheat (*F. esculentum*) subjected to nitrogen treatments (0, 30, 45, 60, and 90 kg N ha^−1^). Different uppercase letters indicate significant differences among treatments according to Tukey’s test (*p* < 0.05). The data are means ± SE (n = 5).

**Table 1 plants-14-02037-t001:** Plant traits in buckwheat (*F. esculentum*) subjected to different nitrogen treatments (0, 30, 45, 60, and 90 kg N ha^−1^).

N Rates (Kg N ha^−1^)	Plant Height (cm)	N° Branches	N° Internode	Shoot Diameter (cm)	Shoot Biomass (g)
0	118.7 ± 1.9 b	5.7 ± 0.5 b	7.9 ± 0.7 b	10.3 ± 1.0 a	10.2 ± 1.1 b
30	120.6 ± 1.7 b	6.3 ± 0.3 b	8.0 ± 0.7 b	11.7 ± 0.6 a	10.5 ± 1.0 b
45	120.67 ± 1.1 b	5.8 ± 0.5 b	6.8 ± 0.6 c	11.7 ± 0.3 a	11.2 ± 1.7 b
60	127.8 ± 1.5 a	7.4 ± 0.7 a	9.8 ± 0.8 a	11.9 ± 0.2 a	13.3 ± 0.1 a
90	123.1 ± 1.6 a	7.2 ± 0.5 a	8.9 ± 0.4 a	11.0 ± 0.5 a	13.2 ± 0.3 a

Different lowercase letters indicate significant differences among treatments according to Tukey’s test (*p* ≤ 0.05). The data represents the mean ± SE (n = 10).

**Table 2 plants-14-02037-t002:** Grain yield components in buckwheat (F. esculentum) subjected to different nitrogen treatments (0, 30, 45, 60, and 90 kg N ha^−1^).

N Rates (Kg N ha^−1^)	Grain per Plant (N°)	Yield per Plant (g)	1000-Grain Weight (g)	Crop Yield (Kg ha^−1^)
0	271.3 ± 30.6 c	8.5 ± 0.9 c	34.0 ± 0.2 a	715.1 ± 50.4 b
30	457.5 ± 30.8 a	14.8 ± 0.6 a	35.0 ± 0.3 a	914.1 ± 45.5 a
45	402.2 ± 14.5 a	12.1 ± 1.9 b	34.9 ± 0.5 a	856.1 ± 19.4 a
60	369.9 ± 40.1 b	12.3 ± 0.4 b	33.4 ± 0.4 a	749.6 ± 28.7 b
90	255.5 ± 37.5 c	8.1 ± 1.8 c	33.6 ± 0.2 a	630.8 ± 30.9 c

Different lowercase letters indicate significant differences among treatments according to Tukey’s test (*p* ≤ 0.05). The data represents the mean ± SE (n = 10).

## Data Availability

The raw data supporting the conclusions of this article will be made available by the authors on request. The data are not publicly available due to privacy.

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
