# Peer review of "Physiological Performance and Grain Yield Components of Common Buckwheat (Fagopyrum esculentum Moench) Cultivated Under Different N Rates"

_plants, 2025, doi:10.3390/plants14132037_

Round 1
Reviewer 1 Report (Previous Reviewer 3)
Comments and Suggestions for Authors
The overall article is relatively simple, with poor innovation. The experiment is only based on one-year results, and the reliability is poor, making it unsuitable for publication
Author Response
Dear reviewer,
Thank for you comments. We attached a letter.
Best regards,
Jorge

Reviewer 2 Report (Previous Reviewer 1)
Comments and Suggestions for Authors
1) All points mentioned in the first edition had been modified.
2) Fig.2 and 3, the black color is not so good, grey may be better.
Author Response
Dear reviewer,
Thank for you comments. We attached a letter.
Best regards,
Jorge

Reviewer 3 Report (Previous Reviewer 2)
Comments and Suggestions for Authors
I am satisfied with the changes.
Author Response
Dear reviewer,
Thank for you comments. We attached a letter.
Best regards,
Jorge

Reviewer 4 Report (New Reviewer)
Comments and Suggestions for Authors
To evaluate the N role on plant traits, photosynthetic performance, and grain yield components in buckwheat under field conditions.
- L21 and other lines, please use single quotation marks for all variety names.
- Table 1and other lines, replace dose with rate. “Nitrogen dose” sounds clinical or pharmaceutical. Please stick with “rate” in scientific and Extension contexts related to crop nutrition.
- L148-187, buckwheat is a well-known P-efficient crop because its rich root exudation, enzymatic mobilization of insoluble phosphates, and low internal requirement and efficient uptake. Please review more literature and rewrite this section.
- L189-199, how did you extract the soil nutrients, what extractant was used?
- L201-213, why did you use those five rates? Logically, it should be 0, 15, 30, 60, 90 kg/ha N. Justification is needed.
- L219-228, you don’t need to repeatedly use LI-6400XT. Sounds too redundant.
- The Abstract should clearly state (1) the objective, (2) methods, (3) key results, and (4) conclusions. The current abstract lacks clarity and essential details. Below is an example of a clear and concise abstract: "Nonexchangeable K+ constitutes a slowly available reserve that may significantly influence K+ fertility of soils. Laboratory and greenhouse experiments were conducted to characterize the K+ supply and nonexchangeable K+–release kinetics in 10 calcareous soils using 0.01 M CaCl2 and 0.01 M oxalic acid extractions. Total K+ uptake by wheat (Triticum aestivum L.) grown in the greenhouse was used to measure plant-available K+. The release of K+ was characterized by an initial fast rate followed by a slower rate. The nonlinear relationship in the early stages of the K+ release may be attributed to the edge sites, and release of K+ from interlayer exchange sites may be responsible for the second part of the K+ release. Kinetics of K+ release was described best with power function, which showed the best fit of the four models tested. Parameters of kinetics models in 0.01 M CaCl2 were significantly related to K+ uptake by wheat. Potassium release was also correlated to initial NH4OAc-extractable K+ and to HNO3-extractable K+."
- Statistical Analysis: Needs greater specificity. The current description is generic and could apply to any study. The following example (https://www.nature.com/articles/s41477-024-01797-7) may help you improve this section.
To compare climatic niche differences between exotic and native species, we ran 19 separate linear mixed-effects models (LMMs) testing whether NCMs depended on species status using the ‘lme4’ R package46. Analysis of variance (ANOVA) with Satterthwaite’s method was applied to determine the significance level through the ‘lmerTest’ R package47 (the unit of replication was species). We included (or not, in the case of singularity fit) family as a random intercept term to account for potential taxonomic autocorrelation in the residuals of the models. Furthermore, two covariates (climatic niche breadth and symmetry) were included in these models to explore potential effects of varying sampling effort among continents and geographical barriers. Specifically, climatic breadth is the range from the 1% to the 99% percentile of a species’ distribution at climatic conditions, which was used to control for potential effects of undersampling the true niche size of species. To avoid the influence of extreme records, we therefore did not use a range from minimum to maximum values to represent climatic breadth. Climatic symmetry is the ratio of the climate range from the 1% and 99% percentiles to the median (climate range of the shorter side divided by that of the longer side), which was used to control for the effects of spatial imbalance in sampling (for example, range truncated by coastlines or mountains). The value range of symmetry is 0 (median is at one of the extremes) to 1 (perfect symmetry on either side of the mean). To control for false discovery rate due to multiple comparisons, we applied the Benjamini–Hochberg procedure for each predictor to adjust the P values across all 19 bioclimatic variables using the ‘p.adjust’ R function. Climatic breadth and symmetry had significant effects on the NCMs of some bioclimatic variables but had no effects on those of others (Extended Data Table 2). Overall, these effects were less likely to be lopsided in favour of exotic or native species (Extended Data Table 3).
To predict the dependence of plant invasion severity on bioclimates at the field survey sites, we estimated how suitable the bioclimatic conditions were for native versus exotic species. For each bioclimatic variable, we first estimated how far the lowest value at the field survey sites (for example, for B01, the coldest site) was from the NCM of each species in terms of percentiles (that is, if the average temperature at the coldest site was in the 20th percentile in the distribution of temperatures in which a plant species occurs in the native range, we gave a value of 30 = |50−20|). Then we repeated this for the field survey site with the highest value (for example, for B01, the warmest site). Next, we used the differences in these values as an estimate of the direction and magnitude of the effect of that bioclimatic variable on the performance of that species. For example, if going from the survey site with the lowest to the highest annual mean temperature (B01) would on average bring the survey site temperature 5.5 percentile points closer to the NCMs for exotic species and 4.7 percentile points farther from the NCMs for native species, this predicts that invasions will be more severe in the warmest site compared with the coldest site. Finally, we conducted a similar set of LMMs and ANOVAs to test whether the predicted effects of each bioclimatic variable depended on species status (the unit of replication was species). We also adjusted the P values across all 19 bioclimatic variables using the same procedure. In these models, the potential effects of taxonomic autocorrelation, varying sampling effort among continents and geographical barriers were also controlled.
To estimate the effects of future climate change for the variables that impacted current invasion severity, we conducted a set of ANOVAs and t-tests with predicted future (100 years) average temperature (B01, +3.3 °C) and annual precipitation (B12, +9%)38 along with diurnal temperature variation (B02, +1 °C)34. In these analyses, we compared the differences in values from the NCMs under current conditions for warming at the warmest and coldest sites, increased precipitation at the driest and wettest sites, and increased diurnal variation at the least and most variable sites. For each of these cases, we performed an ANOVA to test whether the predicted effects depended on species status, and t-tests to examine differences from zero (the unit of replication was species). We performed another analysis with each field site as a replicate in which we examined combined effects of changes in all three bioclimatic variables (B01, B02, B12). At each field site, we calculated the three-dimensional Euclidean distance of all 142 plant species from current versus future conditions to their NCMs for each field site. We used a paired t-test to test whether the predicted effects (current − future distance to NCM) depended on species status, and t-tests to examine differences from zero (the unit of replication was field site).
To compare the relative effects of bioclimatic variables on the richness of exotic and native plants at field survey sites, we ran 19 separate LMMs using the ‘lme4’ R package46. We applied ANOVAs with Satterthwaite’s method to determine the significance level of the interactions between bioclimatic variables and species status using the ‘lmerTest’ R package47 (the unit of replication was richness entry). All variables were scaled before including them in the models. We included the random factors ‘community identity nested in city’ to control for city- and community-specific variation and ‘plot identity nested in community’ to control for exotic and native species richness being measured in the same plot. We also adjusted the P values across all 19 bioclimatic variables. The estimates were used to indicate the direction and magnitude of the effects of bioclimatic variables on the richness of exotic and native plants.
To assess how well exotic and native flora are equilibrated with the climates at field sites, we first calculated the means of NCMs involving annual mean temperature (B01) and annual precipitation (B12) for all species in a plot by abundance weighting, and then used the distances from the NCM means to the climates at field sites to indicate the deviation degrees of community and climate. The closer the distance is to zero, the better the community matches the climate. We used multivariate ANOVA (MANOVA) with Pillai method to compare the degrees of deviation from climate (B01 and B12) between exotic and native communities, and separately examined the degrees of deviation in B01 and B12 using paired t-test (the unit of replication was plot). Plots containing only native plants (31 plots) or only exotic plants (4 plots) were not included in these paired tests.
To examine how exotic species replace native species, we ran three separate generalized additive mixed models (GAMMs) with quasi-Poisson zero structure in the ‘mgcv’ R package48 using the ‘gamm’ function (the unit of replication was plot). Sampling site identity was used as a random factor. GAMM applies non-parametric smoothing to explanatory variables and can thus model nonlinear relationships49,50. The reason GAMMs were used is because we found that species richness responds nonlinearly with increasing exotic dominance. Model comparisons based on Akaike information criterion values also show that nonlinearity has better fits than linearity. The log-transformed ratio of exotic to native abundance in each plot was used to indicate the dominance degree of exotic plants. Because four plots did not include native species, it was not possible to calculate the exotic/native abundance ratio, and 31 plots did not include exotic species and could not be log transformed, hence these 35 plots (7.6% of all plots) were excluded from the models.
To decompose the relative contributions of biotic and abiotic factors to exotic and native species diversity, we conducted variation partitioning analysis with the ‘rdacca.hp’ function and RDA method in the ‘rdacca.hp’ R package51 (the unit of replication was plot). The richness of exotic species was explained by four aspects: temperature, precipitation, human activity and propagule pressure. Native species richness was explained by four aspects: temperature, precipitation, human activity and invasion by exotic species. Climatic variables included B01, B02, B03, B06, B11, B12 and B13, which impacted invasion severity (Extended Data Table 2). The variables involving human activities were population density and gross domestic product. Propagule pressure for exotic species was expressed as the distance from the field site to the estuary. The impact of invasions on native species was represented by exotic species richness. All variables were scale transformed. The relative effect of estimates is the percentage of the variance of each group variables that accounts for the variance of all four groups.
- Conclusion: Should restate the research problem, summarize key findings, and discuss broader implications. It must align more closely with the objectives. Below is an example of a well-structured research paper conclusion: “While the role of cattle in climate change is by now common knowledge, countries like the Netherlands continually fail to confront this issue with the urgency it deserves. The evidence is clear: To create a truly future-proof agricultural sector, Dutch farmers must be incentivized to transition from livestock farming to sustainable vegetable farming. As well as dramatically lowering emissions, plant-based agriculture, if approached in the right way, can produce more food with less land, providing opportunities for nature regeneration areas that will themselves contribute to climate targets. Although this approach would have economic ramifications, from a long-term perspective, it would represent a significant step towards a more sustainable and resilient national economy. Transitioning to sustainable vegetable farming will make the Netherlands greener and healthier, setting an example for other European governments. Farmers, policymakers, and consumers must focus on the future, not just on their own short-term interests, and work to implement this transition now.”
Author Response
Dear reviewer,
Thank for you comments. We attached a letter.
Best regards,
Jorge

Reviewer 5 Report (New Reviewer)
Comments and Suggestions for Authors
Dear Editor,
in this manuscript, several nitrogen treatments were used to test the influence on buckwheat cv grown in the field. In the literature the results of the studies conducted to determine the most appropriate nitrogen dose and its influence on the growth performance of buckwheat plants are variable. In some works low doses of N were the best while in others high doses were preferable or the presence of nitrogen did not influence the characteristics and yields of buckwheat. However, all authors agree that it is necessary to consider the soil and climatic conditions of the cultivation site and the cultivar. The authors operated in specific territories and with a cv that is most likely the most suitable for those conditions.
The article is well written the introduction is exhaustive and the discussion is coherent with respect to the experimental results. In particular, the results are very interesting, these can be helpful not only for researchers and academics, but also for stakeholders operating in certain conditions. The purpose of the study is clear, and the conclusions are credible and interesting. The paper is generally well written and structured. I think that this manuscript is suitable for publication.
Author Response
Dear reviewer,
Thank for you comments. We attached a letter.
Best regards,
Jorge

Round 2
Reviewer 4 Report (New Reviewer)
Comments and Suggestions for Authors
The stat analysis section still needs some work. It is still generic but repeating.
Author Response
Dear reviewer,
Thank you very much for your comment.
We attached a letter.
Best regards.
Jorge

This manuscript is a resubmission of an earlier submission. The following is a list of the peer review reports and author responses from that submission.
Round 1
Reviewer 1 Report
Comments and Suggestions for Authors
1) please compare buckwheat yield of other studies, especially with N dose experiment.
2) please discussed the soil available P on buckwheat yield, as the used soil was relatively low in soil available P.
3) in the conclusion, please state cearly the experimental condition under which your result obtained, that high SOM and N of the experimental soil, low P and low P fertilizer.
Author Response
Dear reviewer,
See the attached letter.

Reviewer 2 Report
Comments and Suggestions for Authors
The abstract must address the need for project .
What was the importance of Buckwheat? How its cultivation can be compared to maize and wheat?
The hypothesis and objectives need to be clearly defined.
What are the broader benefits of this research?
What are the potential future implications of the study?
Why was buckwheat chosen for this research?
What is the recommended fertilizer dosage for the crop, and what is the rationale behind selecting these specific levels?
Discussion needs improvement with some more recent references.
Future implications must be explicitly addressed.
While the study provides valuable insights, there is still some uncertainty regarding its applicability to a wider audience or large-scale field implementation.
Author Response
Dear reviewer,
See the attached letter.

Reviewer 3 Report
Comments and Suggestions for Authors
I have a lot of doubts about the results of the experiment
- Line 100-107 stomatal conductance (gs) and transpiration was reduced by 11% in 30 kg N ha-1 compared to 0 kg N ha-1 (Fig. 3B). By contrast, no changes were observed in gs among 0, 45, 60 and 90 kg N ha-1treatments. Why, I can not understand, Why only the gs and transpiration of 30 kg N ha-1 have been reduced? If the stomatal conductance and transpiration rate decreased, the photosynthetic rate will inevitably decreased, thus affecting growth and development, yet the yield of 30 ha-1 was the highest, which is a contradiction in article.
- Line 122-146 Higher N treatments (60 and 90 kg N ha-1) exhibited greater plant height (about 8%) compared to control plants (Table 1)Likewise, plants under 60 and 90 kg N ha-1 treatments showed the highest number of branches and internode per plant. Why does high N application promoted the growth of plants, yet its yield is the lowest?
Author Response
Dear reviewer,
See the attached letter.

Round 2
Reviewer 1 Report
Comments and Suggestions for Authors
IT is OK for the modificaiton
Reviewer 2 Report
Comments and Suggestions for Authors
The manuscript in the present conditions addresses most of the research questions and applicability , I am satisfied with the changes.